# The Therapeutic Effects of Blueberry-Treated Stem Cell-Derived Extracellular Vesicles in Ischemic Stroke

**DOI:** 10.3390/ijms25126362

**Published:** 2024-06-08

**Authors:** Eunjae Jang, Hee Yu, Eungpil Kim, Jinsu Hwang, Jin Yoo, Jiyun Choi, Han-Seong Jeong, Sujeong Jang

**Affiliations:** 1Department of Physiology, Chonnam National University Medical School, Hwasun-gun 58128, Republic of Korea; jzzahg@naver.com (E.J.); nunuhi0301@gmail.com (H.Y.); wlstn0128@naver.com (J.H.); jiyunchoi10@gmail.com (J.C.); 2Jeonnam Bioindustry Foundation Biopharmaceutical Research Center, Hwasun-gun 58141, Republic of Korea; 3Infrastructure Project Organization for Global Industrialization of Vaccine, Sejong-si 30121, Republic of Korea; eungpil20@hanmail.net; 4Department of Physical Education, Chonnam National University, Gwangju 61186, Republic of Korea; yjin4589@naver.com

**Keywords:** stroke, mesenchymal stem cells, extracellular vesicles, blueberry, microRNA

## Abstract

An ischemic stroke, one of the leading causes of morbidity and mortality, is caused by ischemia and hemorrhage resulting in impeded blood supply to the brain. According to many studies, blueberries have been shown to have a therapeutic effect in a variety of diseases. Therefore, in this study, we investigated whether blueberry-treated mesenchymal stem cell (MSC)-derived extracellular vesicles (B-EVs) have therapeutic effects in in vitro and in vivo stroke models. We isolated the extracellular vesicles using cryo-TEM and characterized the particles and concentrations using NTA. MSC-derived extracellular vesicles (A-EVs) and B-EVs were round with a lipid bilayer structure and a diameter of ~150 nm. In addition, A-EVs and B-EVs were shown to affect angiogenesis, cell cycle, differentiation, DNA repair, inflammation, and neurogenesis following KEGG pathway and GO analyses. We investigated the protective effects of A-EVs and B-EVs against neuronal cell death in oxygen–glucose deprivation (OGD) cells and a middle cerebral artery occlusion (MCAo) animal model. The results showed that the cell viability was increased with EV treatment in HT22 cells. In the animal, the size of the cerebral infarction was decreased, and the behavioral assessment was improved with EV injections. The levels of NeuN and neurofilament heavy chain (NFH)-positive cells were also increased with EV treatment yet decreased in the MCAo group. In addition, the number of apoptotic cells was decreased with EV treatment compared with ischemic animals following TUNEL and Bax/Bcl-2 staining. These data suggested that EVs, especially B-EVs, had a therapeutic effect and could reduce apoptotic cell death after ischemic injury.

## 1. Introduction

A stroke, which is a serious manifestation of cerebrovascular disease, is the first leading cause of death after cancer in the world [1]. Cerebrovascular diseases are divided into hemorrhagic and ischemic strokes according to their pathogenesis [2]. An ischemic stroke is a major cause of death and acquired disorders. Ischemic strokes caused by thromboembolic occlusion of the cerebral artery account for more than 80% of all strokes [3,4]. There are two kinds of ischemic stroke: thrombotic and embolic. A thrombotic stroke, which is the most common disease, occurs when platelet coagulation or blood clots block the blood flow to the brain [5]. An embolic stroke occurs when a temporary or permanent migration is formed in the artery from the embolism that travels to the brain, narrowing the arterial lumen and limiting the blood supply [6]. Acute treatment of strokes is directed at early reperfusion with thrombolysis or hemodynamically through the management of fluid volume and blood pressure [1]. Tissue-plasminogen activator (tPA) is a common drug for urgent patients within six hours of onset, and antiplatelet agents and anticoagulants can be used to prevent a secondary stroke [7,8,9,10,11]. However, there is still a need to discover new drugs for stroke patients.

Stem cell-based therapies hold much promise as potential treatments to repair and regenerate various tissues and organs, enhance plasticity and survival, and restore function in a number of diseases [12,13]. Mesenchymal stem cells (MSCs) are cells present in many tissues and are versatile nonhematopoietic adult cells that can be differentiated into multiple cell types [14]. MSCs can be obtained from a patient’s own tissue, such as adipose tissue, bone marrow, dental pulp, and the umbilical cord [8,15,16,17,18]. The use of MSCs as therapy for stroke would limit immune reactions. However, MSCs cannot cross the blood–brain barrier during intravenous application, highlighting the need for cell-free therapy [12,13].

Extracellular vesicles (EVs), which range from 30 to 200 nm in diameter, have been found to be secreted by most cell types and are detected in almost all bodily fluids in both health and disease conditions [19]. Neurons in the central nervous system (CNS) release EVs, enabling sophisticated communication with astrocytes and microglia [20,21]. This interaction through exosomes supports neuronal regeneration and synaptic function [22]. These macromolecules can cross the blood–brain barrier even though they consist of a variety of proteins, enzymes, transcription factors, lipids, and nucleic acids [19]. EVs are known to play a role in multiple cellular functions, including intracellular communication, cell differentiation and proliferation, angiogenesis, stress responses, and immune signaling [19]. In prior research on strokes, MSC-derived EVs were injected into rodent models and the results showed an improvement in both behavioral outcomes and the infarct area [23,24,25,26]. EVs have also been used in some clinical trials, and it is essential to optimize the purification process for high yield and purity with biological activity. There are various methods for isolating EVs from cell culture media and purification using ultra-filtration, ultra-centrifugation, and density gradients [27]. In the past few decades, multiple studies have demonstrated that MSC-derived EVs can have advantageous effects in various injury and disease states because of their paracrine effects [19,21,22]. MSC-derived EVs have novel beneficial functions characterized by their smaller size, lower complexity, lack of nuclei, increased stability, easier production, longer preservation, and potential as a delivery system of biomolecules [19,28,29,30,31,32,33,34,35,36]. Thus, a scalable method of producing MSC-derived EVs at the laboratory level is needed.

Blueberries contain phytochemicals, including abundant anthocyanin [37,38], and have numerous anti-glycation and neuroprotective effects [39,40]. Numerous studies have discovered beneficial compounds in this fruit, which contains more than 10 bioactive compounds such as peonidin, chlorogenic acid, anthocyanidin, delphinidin, malvidin, and flavonoid. Anthocyanin, which is enriched in berries, can reduce apoptosis by regulating Bcl-2 family proteins and the NF-κB pathway [41]. In aged rats, blueberry extract improved performance in spatial working memory tasks through activation of the ERK/cAMP and CREB/BDNF pathway for daily dietary supplementation [42,43]. Moreover, anthocyanins reduce the damage of hypoxia and ischemic strokes [44,45]. Many studies on blueberries show anti-inflammatory effects on vascular diseases [46,47,48]. According to these studies, the therapeutic effects of blueberries inhibit TNF-a, IL6, and TRL4, which control and regulate inflammatory actions [49,50,51]. The beneficial effects of blueberry extract on ischemic strokes could play important roles in brain–gut signaling to modulate intestinal microbiome metabolism in the rat middle cerebral artery occlusion (MCAo) model [52]. In clinical trials, blueberry, which is one of the polyphenols, has a beneficial function such as reducing blood pressure, preventing a thrombosis, regulating levels of C-reactive proteins, and inhibiting platelet formation [53,54,55,56,57,58,59,60]. However, there is no evidence for the therapeutic effect of blueberry-treated stem cell-derived EVs in ischemic conditions.

In this study, we demonstrated the production and characterization of EVs using a scalable method. To assess the effect of blueberry-treated MSC-derived EVs (B-EVs) compared to untreated MSC-derived EVs (A-EVs) on cells during hypoxic, stroke-like conditions, we performed gene and pathway analyses. In addition, we assessed the therapeutic effect of B-EVs in both oxygen–glucose deprivations (OGD) (in vitro) and MCAo rat models (in vivo) via the regulation of anti-inflammatory effects and anti-apoptotic pathways. The middle cerebral artery (MCA) is a crucial blood vessel in the human brain, supplying blood to a substantial part of the cerebral cortex and the basal ganglia. The MCA is the artery that is most frequently blocked during strokes in humans [61,62]. In current research, the intravascular filament occlusion of the middle cerebral artery (MCA) in rodents stands as the most common model for studying focal brain ischemia [63]. We expected the beneficial effects of blueberry-treated EVs to regulate the anti-apoptotic cell death and reduce the inflammatory factors rather than the not-treated ones.

## 2. Results

### 2.1. Characteristics of EVs

To characterize the EVs derived from MSCs, we determined the size and number of isolated EVs using cryo-TEM and NTA. The TEM analysis showed that both the A-EVs and B-EVs were round with lipid bilayer structure (Figure 1A,B). To increase the production of EVs, we used high-capacity cell factories and large-scale isolation of EVs. According to the NTA, the concentration of A-EVs was determined as 1.74 × 10^8^ ± 5.38 × 10^6^ particles/mL. Interestingly, blueberry treatment increased the concentration of EVs by ten-fold; 6.45 × 10^8^ ± 1.37 × 10^7^ particles/mL. In addition, both A-EVs and B-EVs had an average size of ~150 nm (Figure 1C,D, and Table 1). NTA results showed the size distribution of particles in a liquid suspension with diameters ranging from about 10 to 1000 nanometers (nm) so that the multiple peaks suggested a diverse range of particle sizes. Following a cryo-TEM analysis, we observed the particle size distribution and the particles as a movement (Figure 1E,F).

### 2.2. Cluster Analysis of the EVs

To discover the cellular responses to the EVs, an enrichment analysis of the expressions of the DEG expressions across A-EVs and B-EVs was performed using the Ex-DEGA and MEV. The expression of these genes in A-EVs and B-EVs was shown in a heat map (Figure 2A). B-EVs were the most significantly enriched gene of the DEGs in all specificity index 2.0 foldchange and log2. A total of 207 genes were specifically identified between A-EVs and B-EVs. Fifty-two genes were found to be over-expressed in B-EVs in preparation for A-EVs (Table 2). To demonstrate the over-expressed genes in B-EVs compared to A-EVs, we selected a few target genes and determined the level of gene expression. Following the qPCR, the gene expression in B-EVs was shown: SUMO2, 5.8 ± 0.04; FN1, 3.13 ± 0.1; THBS1, 4.95 ± 0.02; CYTB, 2.84 ± 0.01; NEAT1, 2.01 ± 0.04; and PRDX1, 5.74 ± 0.27 (mean ± standard error, ^###^
*p* < 0.001, ^##^
*p* < 0.01 compared to the A-EVs). This was normalized with the expression in the A-EVs as one (Appendix A). Genes modulated by A-EVs belong to the following functional categories: angiogenesis, cell cycle, death, differentiation, migration, DNA repair, extracellular matrix, immune response, inflammation, neurogenesis, RNA slicing, and secretion. The sentence describes how the results of the transcriptome analysis are visually represented. It states that in the visualization, red dots on the left side signify up-regulated genes, red dots on the right side signify down-regulated genes, and gray dots indicate genes that show no change in expression levels (Figure 2B). The difference of expression of genes in A-EVs and B-EVs is shown in Figure 2C.

### 2.3. The Pathway Analysis of EVs

To demonstrate the predictable function of EVs, we performed a KEGG pathway and GO analysis. The KEGG pathway analysis using the sorted terms by *p*-value (^###^
*p* < 0.001, dark blue color) is demonstrated in Figure 3 and Table 3. Among them, six major signal pathways, including the metabolic pathways, oxidative phosphorylation, focal adhesion, regulation of actin cytoskeleton, Alzheimer’s disease, Parkinson’s disease, and Huntington’s disease, were identified.

A functional enrichment analysis was conducted using Ex-DEGA. DEGs were classified into three functional groups: biology process (BP), cellular component (CC), and molecular function (MF). The GO analysis presented the top 10 or 20 GO terms sorted by *p*-value (^###^
*p* < 0.001). According to the GO terms BF analysis, the upregulated DEGs were mainly related to the construction of extracellular matrix components and inflammatory cytokine activation. Appendix A show that upregulated DEGs were involved in three main functions: (1) construction of translational initiation, protein targeting ER, and protein targeting membrane (see Appendix A), (2) cell-substrate junction and protein focal adhesion (see Appendix A), and (3) structural constituent of ribosome, translation regulator activity, and cadherin binding (see Appendix A).

### 2.4. The Protective Effect of EVs under In Vitro Ischemic Conditions

Studies investigating the use of exosomes have been carried out across a spectrum of diseases, demonstrating their therapeutic potential in conditions such as anticancer effects and ischemic stroke. These findings were used to determine the appropriate concentration of exosomes [64,65]. To assess the effects of EVs in ischemia, we cultured HT22 cells and treated them with A-EVs and B-EVs (1 × 10^10^ particles/mL) under OGD. Under OGD, the number of cells was reduced, and the morphology was comparable to apoptotic-like cells (Figure 4A). The cell viability was also significantly decreased in the OGD group (36.28 ± 1.867%) compared to the control (100%). After A-EV treatments, the viability was increased up to 57.29 ± 2.333% with 1 × 10^9^ particles/mL and 55.02 ± 3.275% with 1 × 10^10^ particles/mL of A-EVs compared with the OGD group (*** *p* < 0.001 and **** *p* < 0.0001, respectively, Figure 4B). There was no difference in the neuroprotective effects depending on the dose of treatment of A-EVs both of 1 × 10^9^ and 1 × 10^10^ particles/mL. Interestingly, the viability was dose-dependently increased after B-EV treatment from 55.03 ± 2.014% with 1 × 10^9^ particles/mL to 60.43 ± 2.703% with 1 × 10^10^ particles/mL. Both A-EVs and B-EVs showed a protective effect on cell death. For the next in vivo study, we chose the concentration of 1 × 10^10^ particles/mL EVs, which revealed a dose-dependent effect on the successful recovery of HT22 cells following 24 h of OGD treatment.

### 2.5. The Protective Effects of B-EVs In Vivo Ischemic Conditions

To study the effect of EVs under in vivo ischemic conditions, we injected 1 × 10^10^ particles/mL of A-EVs or B-EVs 3 days after MCAo. Following our OGD results, we chose the concentration of EVs, which has a potential effect to protect against cell death in OGD conditions, for the next in vivo study. As shown in Figure 5A, the ischemic brain injury was assessed after cerebral infarction in the first and fourth week after injection using MRI. The infarction size was measured as 18.12 ± 0.831% at 1 week and 29.50 ± 3.387% at 4 weeks in the MCAo animals. A total of 4 weeks later, the infarction size was significantly decreased (29.50 ± 3.387% in the MCAo animals, 15.04 ± 1.074% (** *p* < 0.01) in the A-EV-treated animals, and 10.93 ± 1.286% (*** *p* < 0.001) in the B-EV-treated animals (Figure 5B).

To further demonstrate the therapeutic effects of EVs in strokes, we assessed the neurological function in rats after ischemia using cylinder tests every week until four weeks after surgery. There was a significant improvement in right forelimb usage in both the A-EV and B-EV groups over four weeks compared to the MCAo group (Figure 5C). The behavioral results showed that the treatment of EVs had enhanced use of their forelimbs 1 week after surgery (21.62 ± 4.389% in the MCAo animals; 43.28 ± 2.575%, **** *p* < 0.0001 in the A-EV animals; 38.92 ± 1.292%, *** *p* < 0.001 in the B-EV animals) and 2 weeks after surgery (19.77 ± 2.174% in the MCAo animals; 44.78 ± 1.313%, **** *p* < 0.0001 in the A-EV animals; 41.82 ± 3.823%, **** *p* < 0.0001 in the B-EV animals). The measurement of forelimb use was unaffected by the injury and peaked at 3 weeks post-MCAo time point (22.39 ± 1.700% in the MCAo animals; 48.16 ± 3.037%, **** *p* < 0.0001 in the A-EV animals; 47.48 ± 2.199%, and **** *p* < 0.0001 in the B-EV animals). There was a significant difference in behavioral function between the B-EV-treated group and the MCAo group four weeks after the MCAo. At four weeks, the symptoms of the B-EV-treated group improved compared to the MCAo group (24.90 ± 3.637% in the MCAo animals, 45.03 ± 2.424%, *** *p* < 0.001 in the A-EV animals; and 46.93 ± 2.278%, **** *p* < 0.0001 in the B-EV animals). Collectively, these data suggest that B-EVs increased neuronal cell viability after ischemic brain injury both in vitro and in vivo. In addition, A-EV or B-EV treatment did not affect the rat’s body weight compared to the MCAo animals (Figure 5D).

### 2.6. The Effect of EVs on Apoptotic Cell Death

To analyze the increased cell viability of EVs, apoptotic cells in each group were determined using a TUNEL assay. The results showed that the TUNEL-positive cells were increased after the MCAo model and decreased with A-EV and B-EV treatment (Figure 6A). The number of apoptotic cells in each group was 16.00 ± 1.116, 35.00 ± 1.211, 26.90 ± 1.295, and 16.60 ± 1.335 (mean ± standard error, %) in the sham, MCAo, A-EV, and B-EV groups, respectively (**** *p* < 0.0001 in A-EV or B-EV injection compared with the MCAo group, Figure 6B). The apoptotic cells were increased after the induction of ischemia and decreased after injection of A-EVs and B-EVs in stroke animals. These results indicate that EVs could reduce apoptotic cell death after ischemic injury.

### 2.7. The Effect of B-EVs on Neuroprotection

To figure out the neuroprotective effects of EVs, we assessed the survival of neuronal cells with neuron-specific markers, such as NeuN and NFH. In Figure 7, the number of NeuN and NFH-positive cells was decreased after ischemic stroke. However, the neuronal cell marker-positive cells were increased after injection of A-EVs and B-EVs. The percent (relative of sham) of NeuN-positive cells was 46.69 ± 3.964% in the MCAo animals, 179.5 ± 19.70% in the A-EV animals, and 354.7 ± 29.91% in the B-EV animals (**** *p* < 0.0001, compared with the MCAo group, Figure 7A). The percent of NFH-positive cells was 16.92 ± 1.13% in the MCAo animals, 354.7 ± 29.91% in the A-EV animals, and 146.4 ± 14.12% in the B-EV animals. (**** *p* < 0.0001, compared with the MCAo group, Figure 7B). The B-EV-injected group showed more elongated neurofilaments and neurons compared to the other groups.

### 2.8. The Effect of B-EVs on Apoptotic Signaling

Apoptotic cell death was demonstrated by Bax/Bcl-2 immunohistochemistry after four weeks of EV injection (Figure 8A). The number of Bax-positive cells was 51.50 ± 1.211, 40.03 ± 4.128, and 40.04 ± 2.593 (mean ± standard error %) in the MCAo, A-EV, and B-EV groups, respectively (** *p* < 0.01, * *p* < 0.05, compared with the MCAo group). The number of Bax-positive cells was increased with the induction of ischemic injury, while the EV treatment decreased the number of Bax-positive cells in the ischemic rats (Figure 8B).

Further, the level of Bcl-2 expression in the MCAo group was decreased, while EV treatment increased the Bcl-2-positive cells (37.62 ± 3.116 in the A-EV animals and 38.66 ± 1.538 in the B-EV animals) (mean ± standard error %; ** *p* < 0.01, *** *p* < 0.001 compared with the MCAo group, Figure 8C). These results indicate that the EV treatment reduced apoptosis after ischemia.

## 3. Discussion

In this study, we aimed to isolate and characterize EVs from hADSCs with or without blueberries using NTA and Cryo-TEM. The morphology and size distribution of EVs were assessed using Cryo-TEM and NTA. The results showed that both A-EVs and B-EVs had a round shape with a bilayer extracellular vesicle membrane structure and large-scale production. The expression of genes associated with cell death, neurogenesis, and cell cycle were upregulated with A-EVs and B-EVs. According to the GO annotation and KEGG pathway expression profiling analysis, B-EVs have the potential to affect human disease and following cellular responses. In addition, B-EVs were assessed under in vitro OGD and in vivo MCAo ischemic conditions. In this study, we found that infusing B-EVs after cell damage induced by OGD promotes neuronal cell regeneration. We found that B-EV treatment increased the number of cells, improved motor behavioral ability, decreased the infarct area in the brain, reduced the number of TUNEL-positive cells, regulated the apoptotic pathway via Bax and Bcl-2, and significantly increased neuronal cells in the brain. In addition, this study is the first result about the beneficial effects of the blueberry-treated MSC-derived EVs.

EVs penetrate target cells by various endocrine pathways, such as phagocytosis, clathrin- and caveolin-mediated endocytosis, micropinocytosis, and membrane fusion, affecting distant cells as well as the paracrine pathway affecting neighboring cells to communicate with the microenvironment [66,67]. MSC-derived EVs have broad regenerative potential in various human diseases to overcome serious risks with MSC treatment, such as malignant transformation, immune rejection, and embolization [66,68]. However, the secreted EVs from cultured cells are usually less than 1 μg/mL and are insufficient for clinical translation [69]. The International Society for Extracellular Vesicles (ISEV) announced guidelines for EV research to standardize experiments carried out by scientists in different laboratories [66,70]. Ultrafiltration uses membranes with different molecular weights to separate and isolate EVs in a size-dependent manner. TFF is an isolation method that concentrates and filters out particles using a cross-flow filtration principle [71]. This method uses tangential flow across the surface, avoiding filter clogging in a relatively short period of time. The TFF methods of EV isolation led to higher immunomodulation potency of protein and lipids and yielded more EVs than other isolation methods [72,73]. In addition, current methods of isolation are either time consuming or expensive, and no production protocol complies with Good Manufacturing Practices [66]. Thus, cell-free therapy, such as EVs, is necessary to develop a scalable method of producing EVs for clinical needs. The development of methods for large-scale production, isolation, and drug loading is necessary to improve the efficiency and therapeutic potential of EVs [74]. TFF, also known as the cross-flow filtration principle, passes parallel to the filter rather than vertically pushing out the membrane that blocks the filter medium. TFF improves the purification yield of exosomes more than ultracentrifugation and ultrafiltration/diafiltration methods [75].

The KEGG pathway enrichment analysis in the present study demonstrated neurodegenerative diseases and Wnt signaling, necroptosis, and neurotrophin signaling pathways. In the present study, a total of 207 DEGs were identified, including 144 upregulated and 63 downregulated DEGs. A GO enrichment analysis demonstrated that the upregulated genes were mainly associated with the mitochondrial protein complex, ribosomes, cell–substrate junctions, oxidative phosphorylation, antioxidant activity, and translation factor activity, acting on NAD(P)H. Mitochondria are involved in many processes related to cellular function. Mitochondrial functional changes have an important impact on cell function and disease progression [76]. Normal functioning mitochondria are important for nerve survival. Mitochondrial dysfunction is related to several ischemic diseases, including ischemic stroke and myocardial infarction. The oxidative phosphorylation process within the mitochondrial substrate provides the majority of energy to cells [77].

Blueberries provide a promising means to increase EV production by MSCs. Contained in blueberry, flavonoids, and polyphenols have been reported to have therapeutic effects on neurological disease [78,79]. A benefit of blueberry supplements in cardiovascular disease was demonstrated using randomized controlled trials and discovered the regulating of inflammatory factors [51]. The prevention of stroke promoted by polyphenols relies mainly on their effect on ischemic stroke systems including neuroprotective roles in many cellular mechanisms [60]. In rat ischemic stroke models, flavonoids inhibited proinflammatory cytokine production and decreased cerebral infarction [52,80]. Blueberries’ high anthocyanin content may play a role in providing health benefits against chronic diseases [81]. The blueberries used in this study are classified as anthocyanins. Anthocyanins are a type of secondary metabolite in plants, produced through the phenylpropionate and flavonoid synthesis pathways [82]. Blueberries are known to reduce the damage caused by ischemic stroke. Additionally, blueberries have been associated with better cognitive performance, likely due to a reduction in oxidative stress in the brain and an increase in serum antioxidant capacity [48].

In addition, a wild lowbush blueberry had neuroprotective effects on global cerebral ischemia/reperfusion injured rats through downregulation of iNOS/TNF-α and upregulation of miR-146a/miR21 expression [51]. Similarly, when neuronal cells were treated with those genes under OGD conditions, a significant increase in expression was observed in B-EVs. Following the gene expression profiling, B-EV treatment showed that the genes *SUMO2*, *THBS1*, *NEAT1*, *PRDX1*, *FN1*, and *CYTB* were upregulated compared to A-EVs (Appendix A). Those target genes are related to the regulation of apoptosis, protection of neuronal cell death, or regulation of neuronal cell differentiation. SUMO2 is known for its neuroprotective effects [83,84,85] while THBS1 regulates axon growth [86]. CYTB plays a role in the regulation of apoptosis [87], and NEAT1 is linked to decreased neuronal apoptosis [88,89]. PRDX1 and FN1 are involved in the regulation of neuronal cells [90,91,92]. Therefore, MSC-derived exosome has a beneficial effect in protecting against cell death, when the exosome is with blueberry extract rather than without blueberry.

There are two separate areas of ischemic stroke: the ischemic core and the ischemic penumbra [93]. Apoptosis of neurons accounts for a significant proportion of the ischemic penumbra [94]. In general, neurogenesis after ischemic stroke is very challenging [95]. In addition, there are other forms of cell death, such as intrinsic apoptosis, extrinsic apoptosis, necrosis, ferroptosis, and pyroptosis [96]. The intrinsic apoptotic pathway refers to mainly the mitochondrial-mediated pathway [97]. This pathway is mediated by Bax and Bak insertion into the mitochondrial membrane, followed by cytochrome c release into the cytosol from the mitochondrial intermembrane space. The Bcl-2 family is an anti-apoptotic protein family and inhibits the release of cytochrome c [98]. This study’s findings also revealed the Bax expression in the hippocampus significantly declined in the B-EVs treatment groups compared to the MCAo group. However, the level of Bcl-2 was increased with B-EV treatment.

EVs refer to several types of vesicles including, exosomes, microvesicles, and apoptotic bodies [19]. EVs are released by donor cells into the extracellular environment to perform diverse biological functions between a parent cell and surrounding cells. EVs are a promising biological gene delivery system as liposome-mediated drug delivery in the clinical field [19]. EV therapy is emerging as a promising cell-free alternative to traditional cell-based therapies. It aims to avoid complications such as immune rejection, cancer risk, and vascular occlusion while offering advantages in storage and transportation since it doesn’t involve live cells. The successful commercialization of exosome treatments depends on efficient mass production, as well as advanced techniques for separation and purification [99]. To advance to clinical trials, it’s essential to identify an exosome isolation method that demonstrates significant therapeutic benefits for ischemic stroke, establish strict product quality control through validated quantitative and qualitative analyses, and secure regulatory approval [100,101]. Further studies have demonstrated that MSC applications in damaged tissues are the result of paracrine signaling, such as secreted vesicles [102,103,104,105]. Interestingly, EVs secreted by MSCs may replace stem cell therapies in various damaged tissues [19,21,22]. In particular, investigations have revealed that MSC-derived EVs induced repair in mouse models of wound healing and myocardial infarction [30,106,107,108,109] and enhanced tissue regeneration in various contexts, including neurological, respiratory, cartilage, kidney, cardiac, and liver disease [19,28,29,30,31,32,33,34,35,36]. MSC-derived EV therapy is one of the most promising ways of treating ischemic stroke since it aims to inhibit apoptosis and increase neuronal cells. Research has demonstrated that mesenchymal stem cell (MSC)-derived extracellular vesicles exhibited therapeutic benefits in the treatment of ischemic stroke. therapeutic effects of exosomes in mice following ischemic stroke have been associated with long-term neuroprotection and enhanced angiogenesis [110,111]. The interaction between pro-apoptotic and anti-apoptotic Bcl-2 family proteins in the outer mitochondrial membrane plays an important role in neuronal cell survival [112]. This interaction affects the expression of the pro-apoptotic protein Bax and the anti-apoptotic protein Bcl-2 in MCAo and OGD conditions [113].

EVs vary in size, content, surface markers, and origin, making them challenging to isolate. Current methods for isolating and purifying exosomes rely on factors such as size, surface charge, or immunoaffinity. Size-based isolation techniques primarily encompass ultrafiltration and size-exclusion chromatography, which are rapid and well suited for large-scale applications. Increasing exosome production and developing a range of therapeutic effects is crucial for advancing the clinical use of stem cell-derived exosomes, potentially broadening their therapeutic application.

The present study had several limitations other than those aforementioned. First, the extracellular vesicles contain many kinds of components such as miRNA, RNA, DNA, proteins, and apoptotic body. We need to prove the effect of a single molecule like miRNA. We are now working on the miRNA, which is related to cell survival, apoptosis, ER stress, or cell cycle for the next studies. Second, this study did not consider the effect of blueberries directly. There are many studies about the effect of blueberries in protecting against cell death including via an antioxidant effect. So, we did not test the direct effect of blueberry in the stroke or apoptotic cell death. Finally, further prospective study on this topic is necessary, such as research on MSC-EV-derived miRNA or target protein.

Taken together, we established cell and animal models of ischemic stroke and showed that B-EVs increased viability after OGD-induced cell injury, reduced infarct volume, and improved motor behavior impairments. We also suggest that the ultimate goal of B-EV treatment is to reduce cellular apoptosis and improve neurogenesis and motor function. In addition, EVs, especially blueberry-treated MSC-derived EVs, could be a potential drug to rescue neuronal cell death in ischemia such as the brain and spinal cord (Figure 9).

## 4. Materials and Methods

### 4.1. Animals

All experimental procedures were consistent with the Institutional Animal Care and Use Committee (CNU IACUC-H-2022-36) of Chonnam National University Medical School, Hwasun-gun, Republic of Korea. Male Sprague–Dawley rats (180–220 g, 8 weeks old) were purchased from Damul Science (Daejeon, Republic of Korea) and randomly assigned to 3 groups using a random table method: the sham-surgery and sedentary group (Sham, *n* = 8), the ischemia and sedentary group (MCAo, *n* = 8), the ischemia and A-EV-treated group (MCAo + A-EV, *n* = 8), and the ischemia and B-EV-treated group (MCAo + B-EV, *n* = 8). The rats were housed in the animal room at 22–24 °C with a 12 h light/dark circle and free access to food and water.

### 4.2. Cell Culture

#### 4.2.1. Human Adipose Tissue-Derived Stem Cells

Adipose tissues from human donors were obtained according to the guidelines established by the Ethics Committee at the Chonnam National University Medical School (IRB: I-2009-03-016). Human adipose tissue-derived MSCs (hADSCs) were cultured in Dulbecco’s Modified Eagle’s medium (DMEM, Gibco BRL, Grand Island, NY, USA) with 10% fetal bovine serum (FBS, Thermo Fisher Scientific Co., Waltham, MA, USA) and 1% penicillin-streptomycin (Thermo Fisher Scientific Co.) and placed at 37 °C in a humidified incubator [114,115] (5% CO_2_/95% air, N-Biotech, Bucheon-si, Gyeonggi-do, Republic of Korea). The cells were grown to approximately 80–90% confluence and then cultured and isolated for the next experiments.

#### 4.2.2. HT22 Cells

The mouse hippocampal neuronal cell line HT22, which was derived from the immortalization of mouse neuronal tissues, was purchased from Merck (Darmstadt, IN, USA). The cells were cultured in DMEM with 10% FBS and 1% penicillin-streptomycin and placed at 37 °C in a humidified incubator (5% CO_2_/95% air) [116]. When the cells were grown to approximately 70% confluence, the cells were dissociated for the experiments.

### 4.3. EV Isolation and Characterization

#### 4.3.1. Blueberry Extract

A total of 50 g of dried and ground blueberries was added to 1 L of 100% water or ethanol solvent and refluxed at 250 °C for 3 h. After cooling, the mixture was centrifuged to remove sediment, and impurities were filtered out to obtain a filtrate. The filtrate was then concentrated under reduced pressure at 50 °C and 750 mmHg. Finally, the concentrate was freeze-dried at −80 °C to obtain the dry extract.

#### 4.3.2. Blueberry Treatment and EV Isolation

For the massive production of EVs, hADSCs were placed into a cell factory system (Thermo Fisher Scientific Co.) and cultured until approximately 50% confluency. The cells were then cultured for 48 h in serum-free medium (SFM, Xcell therapeutics, Seoul, Republic of Korea) containing 1% penicillin-streptomycin or 1% Glutamax (Gibco BRL) and incubated with blueberry extract (100 μg/mL), which was provided by the Division of Food and Nutrition, College of Human Ecology, Chonnam National University, for the next 24 h. The A-EVs and B-EVs were isolated from conditioned medium (1000 mL) through a tangent flow filtration (TFF, Repligen, Waltham, MA, USA) system using a weight cutoff-based 300 kDa membrane. The TFF system, compared to conventional isolation methods, offers higher isolation yield and purity. First, to remove large particles, the supernatant was harvested and filtered with a 0.22 μm polyethersulfone membrane filter (Repligen). For EV purification, the KrosFlo KR2i TFF system (SpectrumLabs, Los Angeles, CA, USA) equipped with modified polyethersulfone (mPES) hollow fiber filters with a membrane pore size of 300 kDa (MidiKros, 115 cm^2^ surface area, SpectrumLabs) at a flow rate of 100 mL/min and a tubing size of 16 was used [71].

#### 4.3.3. Particle Size and Concentration Analysis

Nanoparticle tracking analysis (NTA) is a widely used method for simultaneously measuring both size distribution and particle concentration [117,118]. Total EV protein concentration was determined using the Pierce BCA protein assay kit (Thermo Scientific, Altrincham, UK). A nanoparticle tracking analysis (NTA) was performed on EV samples to determine particle size and concentration using NanoSight LS300 (Malvern Instruments Ltd., Malvern, UK). The EV samples were diluted 1:100 in PBS and injected into the NanoSight LS300 with the camera level (brightness of the image achieved with a combination of shutter and gain) set to 13, and 4 videos of 30 s were acquired for each sample. Particle size and concentration were determined with NTA 3.0 software (Malvern Instruments Ltd.).

#### 4.3.4. Transmission Electron Microscopy (TEM)

A-EVs and B-EVs were analyzed using a Tecnai F20 cryo-transmission electron microscope (cryo-TEM, Tecnai F20, Thermo Fisher Science, Waltham, MA, USA) at the Microscope Facility of Korea Institute of Science and Technology (Daejeon, Republic of Korea).

### 4.4. EV microRNA Isolation and Analysis

Total EV RNA was isolated using the Qiagen miRNeasy Mini Kit (Qiagen, Manchester, UK) according to the manufacturer’s protocol. Global expression patterns of EV miRNAs were examined using a microarray chip containing 1917 probes for homo sapiens microRNAs (miR-Base 22). Differentially expressed microRNAs were defined by a threshold of *p* < 0.05 and a fold change of > 2.0. Significantly altered microRNAs were further analyzed to predict their target genes and pathways.

### 4.5. Bioinformatics Analysis

To characterize the biological function of EVs, we analyzed gene ontology (GO) to explore the functional role of EV target genes in terms of biological processes. A GO enrichment analysis of differentially expressed genes (DEGs) was implemented by the cluster Profiler R package, in which gene length bias was corrected. This analysis used a biological system defined by the Kyoto Encyclopedia of Genes and Genomes (KEGG) pathways (http://mirwalk.umm.uni-heidelberg.de/ accessed on 14 February 2022) [119].

### 4.6. Gene Expression Profiling

Gene array and data analysis were carried out by E-BIOGEN Inc. (Seoul, Republic of Korea). Excel-based Differentially Expressed Gene Analysis (ExDEGA) software 5.0 (E-BIOGEN Inc., Seoul, Republic of Korea) was used with the multi-average method and DEG analysis DEGs. Heatmap generation and clustering analyses were performed using MeV (version 4.9.0) for selected genes.

### 4.7. Oxygen–Glucose Deprivation

To initiate oxygen–glucose deprivation (OGD), HT22 cells were maintained in a glucose-free medium (Gibco BRL) and stimulated with 5% CO_2_ and 95% N_2_ in a humidified chamber for 10 h. Then, the cells were exposed to 5% CO_2_ and 95% O_2_ for another 24 h, modified from a previous study [120].

### 4.8. Detection of Cell Viability

To demonstrate the effect of EVs on cell viability, HT22 cells were treated with 1 × 10^9^ particles/mL or 1 × 10^10^ particles/mL of EVs for 24 h. Cells were trypsinized and combined with trypan blue to assay cell viability. To measure the cell viability, we randomly counted and measured the surviving cells using the LUNA-IITM (Logos Biosystems, Anyang, Republic of Korea) automated cell counter in triplicate experiments. In addition, we confirmed the surviving or dead cells under a microscope (Olympus, Tokyo, Japan).

### 4.9. Animal Model of Ischemic Stroke

Cerebral ischemia was introduced by occluding the left middle cerebral artery (MCA) permanently, as described in previous studies [121,122]. Briefly, the rats were anesthetized with 50 mg/kg pentothal sodium (JW Pharmaceutical, Seoul, Republic of Korea) using intraperitoneal injection and maintained under isoflurane USP (Troikaa Pharmaceuticals Ltd., Ahmedabad, India) during surgery. To induce MCAo, the muscle layer was separated to expose the common carotid artery (CCA), the external carotid artery (ECA), and the internal carotid artery (ICA). A silicone-coated nylon monofilament (diameter 0.23 mm, L.D-403556PK10, Doccol corporation, Sharon, MA, USA) was inserted from the ECA into the ICA lumen, and a 15–19 mm lumen thread blocked the origin of the MCA. After an hour, the occluded filament was removed, and the slip knot on the CCA was opened for blood reperfusion [123]. The rats with no evidence of acute neurological deficits or with evidence of hemorrhage were excluded from the analysis. Sham-operated animals were handled the same way, exposing the CCA but no occlusion was performed. To inject the EVs, the animals were anesthetized and placed on a stereotaxic apparatus. After 3 days of MCAo, a total of 1 × 10^10^ particles/mL of B-EVs was administered by intracerebroventricular injection (ICV). All the experimental procedures for the animals are described in Appendix A.

### 4.10. Cylinder Test

To investigate the behavior of the animals, all the animals were tested using both forelimbs inside of the cylinder before starting the stroke. The functional behavior of the animals (*n* = 8 for each group) was tested weekly for four weeks. To detect asymmetric use of the forelimbs, animals were placed in cylinders (21 cm diameter × 40 cm height), and the number of contacts of the forelimbs (touching the wall using the front paws) was recorded for 10 min [124,125,126]. The use of the right (R) versus left (L) forelimb is calculated as a percentage of total contacts: R/(R + L) × 100 [127].

### 4.11. Magnetic Resonance Imaging (MRI)

During MRI, all rats were anesthetized using isoflurane and maintained to minimize head motion. T2-weighted data for template construction and functional data of animals were acquired on a 7.0 T animal MRI scanner (M7 Compact MRI system, Aspect Imaging, Shoham, Israel). In the T2-weighted MRI, the volume of the infarct area was measured and visualized using Infanview software version 4.67 (https://www.irfanview.com/ (accessed on 12 October 2023); Austria, Europe).

### 4.12. Terminal Deoxynucleotidyl Transferase dUTP Nick end Labeling (TUNEL) Assay

To investigate whether the EVs had any effect on apoptosis, a TUNEL assay was performed using the DeadEnd™ Colorimetric TUNEL System (Promega, Fitchburg, WI, USA) according to the manufacturer’s protocol. Briefly, cryo-sections were fixed by immersing the slides in 4% paraformaldehyde (PFA, Biosesang, Yongin-si, Gyeonggi-do, Republic of Korea) for 15 min followed by washing with PBS. Sections were permeabilized with proteinase K solution for 20 min after an equilibration buffer for 10 min. The sections were immersed in a TdT reaction mixture buffer and incubated at 37 °C in a humidity chamber for 60 min. Finally, the sections were stained with a daminobenzidine (DAB, Enzo, New York, NY, USA) solution. All images were taken under a microscope (ZEISS Axio Vert.A1, Göttingen, Germany).

### 4.13. Immunohistochemistry

For immunohistochemical staining, sections were fixed and blocked in 10% normal goat serum (NGS, Vector Laboratory, Newark, CA, USA) for 30 min at room temperature. The slides were incubated with primary antibodies (Bcl-2 and Bax, Santa Cruz, TX, USA) for 120 min at room temperature and then with a secondary antibody (HRP-conjugated IgG; Santa Cruz, TX, USA) for 60 min. The immunoreaction was detected by peroxidase-conjugated kits (Vector Laboratory, Newark, CA, USA) for 30 min and visualized with DAB [128].

### 4.14. Immunofluorescence

At four weeks after model establishment, each group was perfused, and the brain tissue was isolated and post-fixed in 4% PFA. Serial sections of the brain were cut at 15 μm thickness on a cryostat (Leica, Wetzlar, Germany) and incubated with primary antibodies, neuronal nuclei (NeuN, Cell Signaling, Danvers, MA, USA), and neurofilament-heavy chain (NFH). After, secondary antibodies (Alexa-488 and Alexa-594-conjugated antibodies, Cell Signaling Technologies) were used for fluorescent visualization. 4′, 6-diamidino-2-phenylindole (DAPI, Life Technology, Carlsbad, CA, USA) was used to label the nuclei.

### 4.15. Statistical Analysis

Statistical analyses were conducted using GraphPad Prism 5 (GraphPad Software, La Jolla, CA, USA). To assess the significance of differences between two groups, such as A-EVs versus (vs.) B-EVs, control vs. MCAo in vivo, or control vs. OGD in vitro, a two-tailed unpaired Student *t*-test was used. A *p*-value below 0.05 was considered to be statistically significant, and the level of significance is indicated ^###^
*p* < 0.001, ^##^
*p* < 0.01, ^#^
*p* < 0.05. For comparisons among three or more groups, such as MCAo vs. A-EVs vs. B-EVs in vivo or OGD vs. A-EVs vs. B-EVs in vitro, one-way ANOVA was applied, followed by the Tukey post hoc test. For multiple comparisons, two-way ANOVA with the Bonferroni post hoc test was performed. A *p*-value less than 0.05 was considered statistically significant. A *p*-value below 0.05 was considered to be statistically significant, and the level of significance is indicated **** *p* < 0.0001, *** *p* < 0.001, ** *p* < 0.01, * *p* < 0.05.

## Figures and Tables

**Figure 1 ijms-25-06362-f001:**
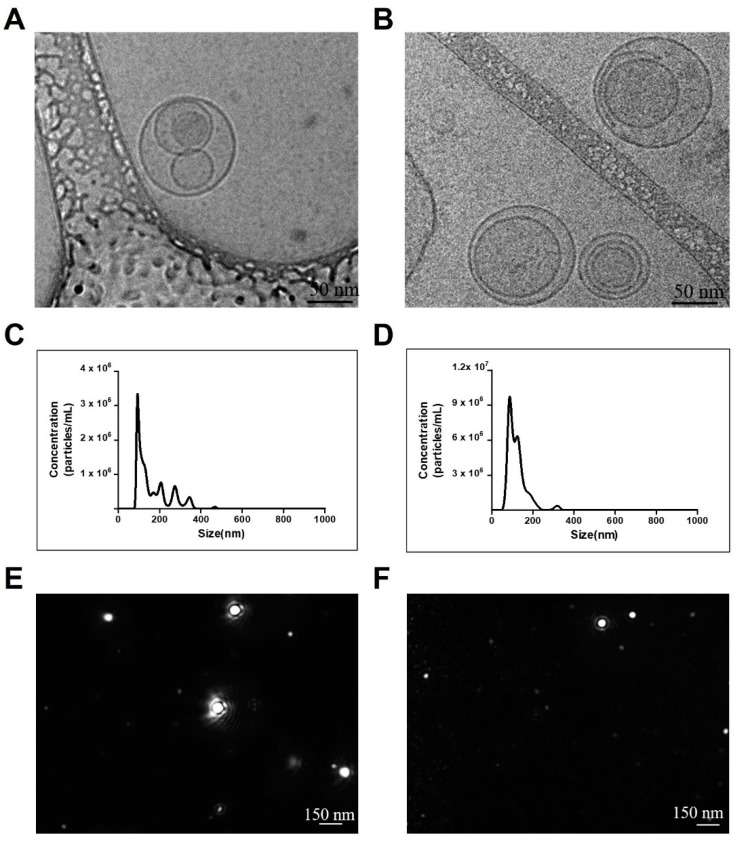
Characterization of EVs isolated from hADSCs and blueberry-treated hADSCs. TEM images of A-EVs (**A**) and B-EVs (**B**) with a round shape with a lipid bilayer structure. Scale bar indicates 50 nm. NTA results show the particle size and concentration of the A-EVs (**C**) and B-EVs (**D**). (**E**,**F**) show the snapshot of A-EVs and B-EVs during analysis, respectively. Scale bar indicates 150 nm. All experiments were repeated at least three times.

**Figure 2 ijms-25-06362-f002:**
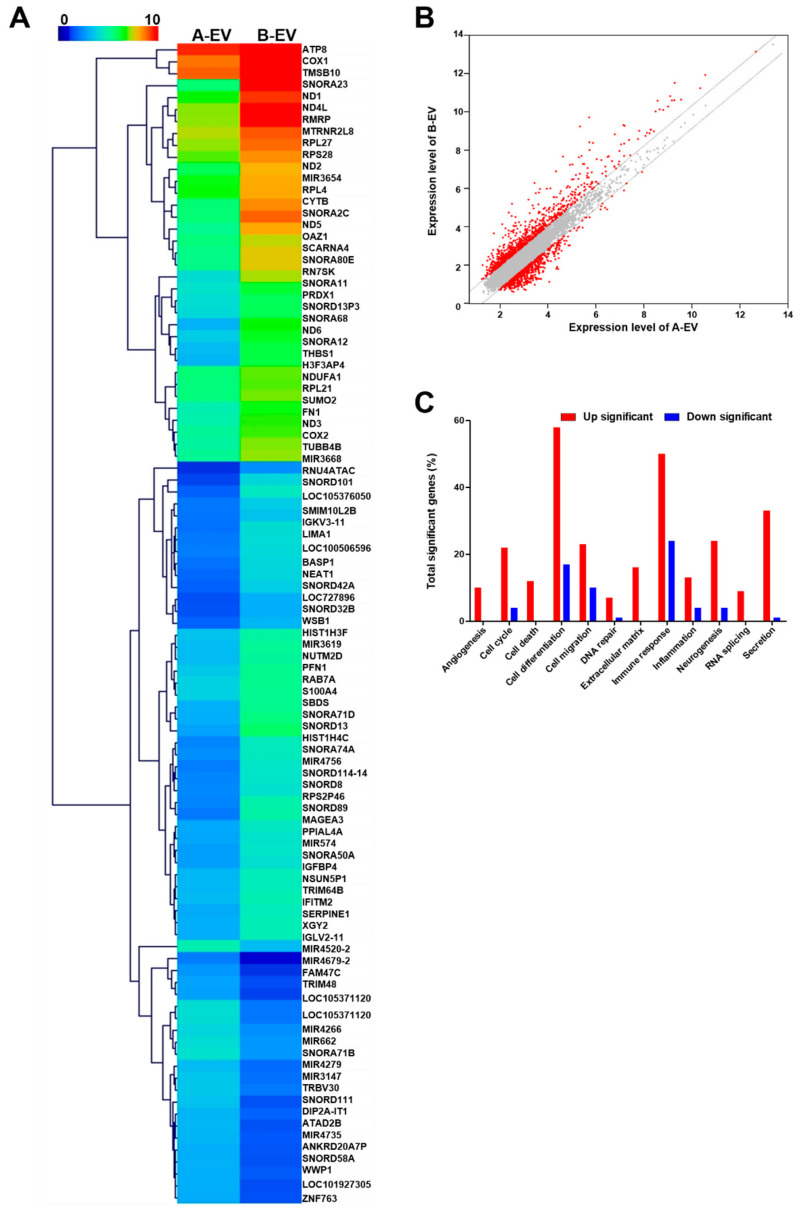
Gene expression profiling of A-EVs and B-EVs. (**A**) Heat map of significantly upregulated genes in B-EVs compared to A-EVs. (**B**) The scatter plot of differentially expressed genes (DEGs) screened by fold change and average signal intensity (log2) according to RNA-sequencing results comparing A-EVs and B-EVs. The X-axis displays the average signal intensity (log2) of genes in A-EVs and the Y-axis corresponds to the average signal intensity (log2) of genes in B-EVs. Red dots indicate up- or down-regulated genes in the B-EV compared to A-EV and gray dots indicate no changes between A-EV and B-EV. (**C**) The distribution graph for the populations of genes A-EVs and B-EVs.

**Figure 3 ijms-25-06362-f003:**
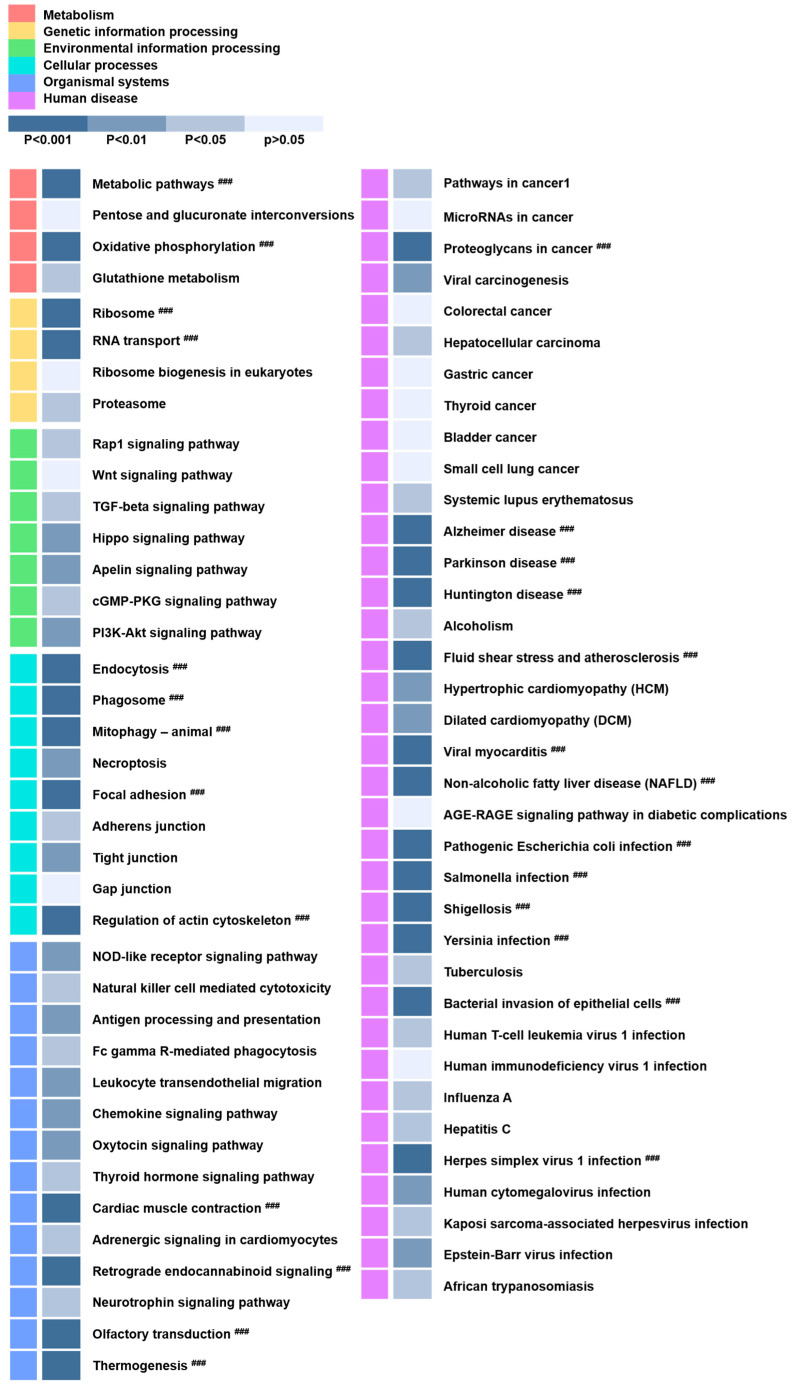
The results of the KEGG pathway analysis. The six KEGG classifications are displayed between A-EVs and B-EVs in different colors (^###^
*p* < 0.001, dark blue color).

**Figure 4 ijms-25-06362-f004:**
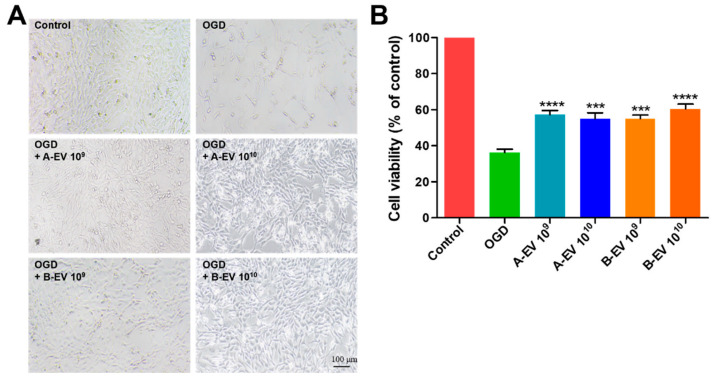
Cell viability of EV treatment under OGD conditions. (**A**) HT-22 cells were observed by microscopy after treatment with A-EVs of B-EVs. A-EVs and B-EVs alleviated OGD-induced decreases in cell viability and apoptosis. (**B**) The cell viability was determined with A-EV or B-EV treatment in OGD cells. All experiments were repeated at least three times. *** *p* < 0.001 and **** *p* < 0.0001 compared with the OGD group.

**Figure 5 ijms-25-06362-f005:**
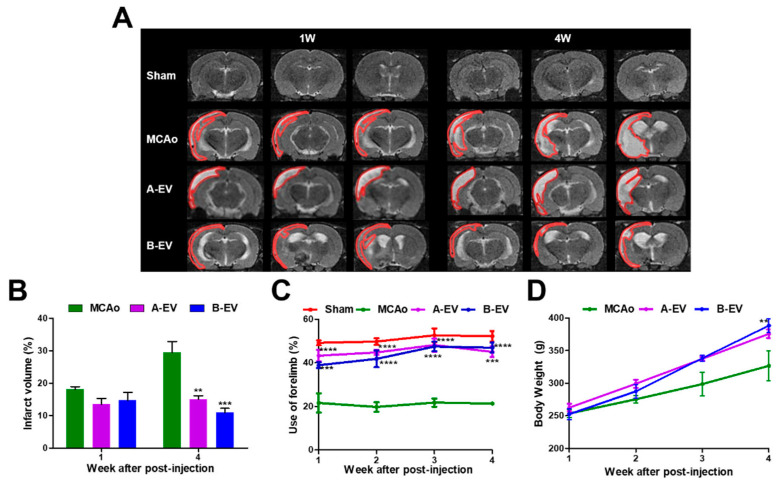
The protective effects of A-EVs and B-EVs in an in vivo ischemic model. (**A**) T2 MRI images of each group were taken in the first and fourth week after surgery. The infarct areas are represented with red lines. Scale bar indicates 10 μm. (**B**) Measurement of infarct volume was observed at one and four weeks after surgery. (**C**) Behavioral tests were conducted every week until four weeks. (**D**) Determination of bodyweight for four weeks. All experiments were repeated at least three times. ** *p* < 0.01, *** *p* < 0.001, and **** *p* < 0.0001 compared with the MCAo group.

**Figure 6 ijms-25-06362-f006:**
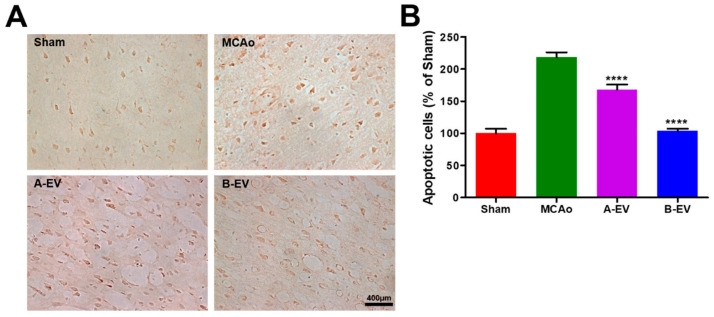
Determination of the number of apoptotic cells. To evaluate the reduced apoptotic cells after injection of A-EVs and B-EVs, TUNEL staining was conducted. (**A**) Apoptotic cells were stained with brown color to indicate TUNEL-positive cells. (**B**) We quantified the number of TUNEL-positive cells. The scale bar indicates 400 μm. All experiments were repeated at least three times. **** *p* < 0.0001, compared with the MCAo group.

**Figure 7 ijms-25-06362-f007:**
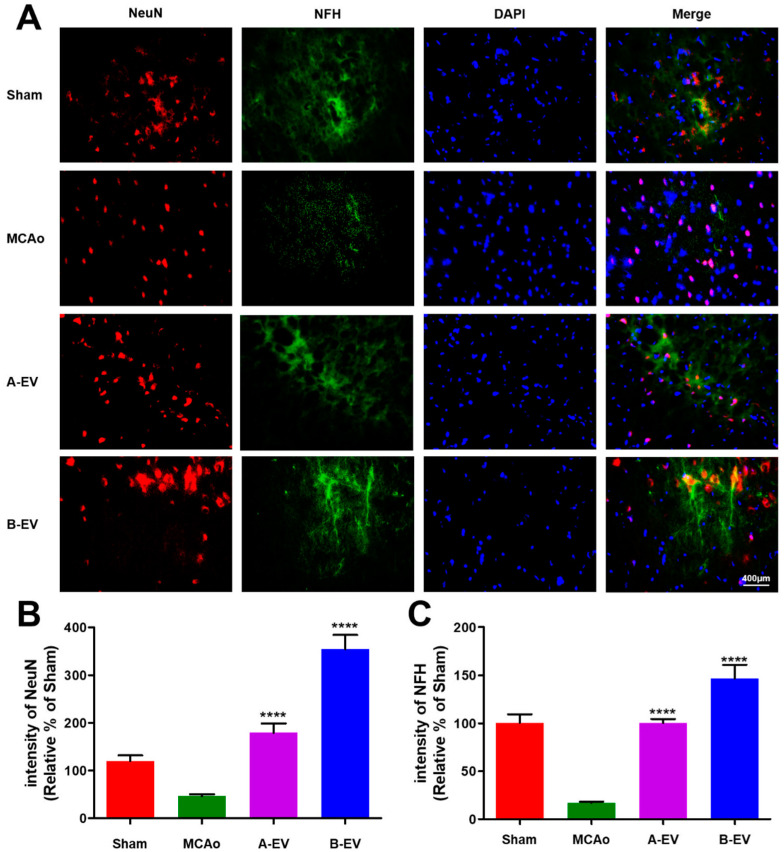
Fluorescence immunohistochemistry of neurons after ischemia and injection of EVs. (**A**) NeuN-positive cells (red) and NFH-positive cells (green) were highly expressed following EV treatment compared to the MCAo animals. DAPI (blue) was used to visualize the nuclei. The scale bar indicates 400 μm. The number of positive cells of NeuN (**B**) and NFH (**C**) is shown. The ratio of positive cells to nuclei was calculated for each group. All experiments were repeated at least three times. **** *p* < 0.0001 compared with the MCAo group.

**Figure 8 ijms-25-06362-f008:**
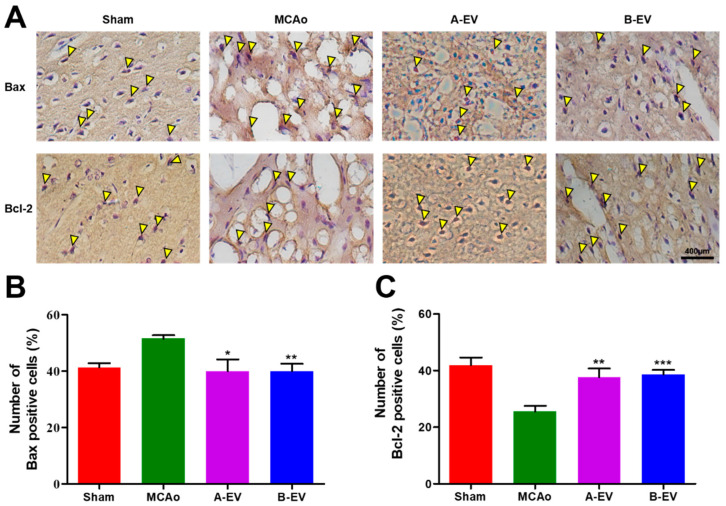
Immunohistochemical assay for Bax and Bcl-2. Apoptosis was detected by DAB staining (**A**) and the number of Bax-positive (**B**) and Bcl-2-positive (**C**) cells was calculated after four weeks of EV injection. Yellow triangle indicates the DAB-stained (Bax- or Bcl-2-positive) cells. The scale bar indicates 400 μm. All experiments were repeated at least three times. *** *p* < 0.001, ** *p* < 0.01, and * *p* < 0.05 compared with the MCAo group.

**Figure 9 ijms-25-06362-f009:**
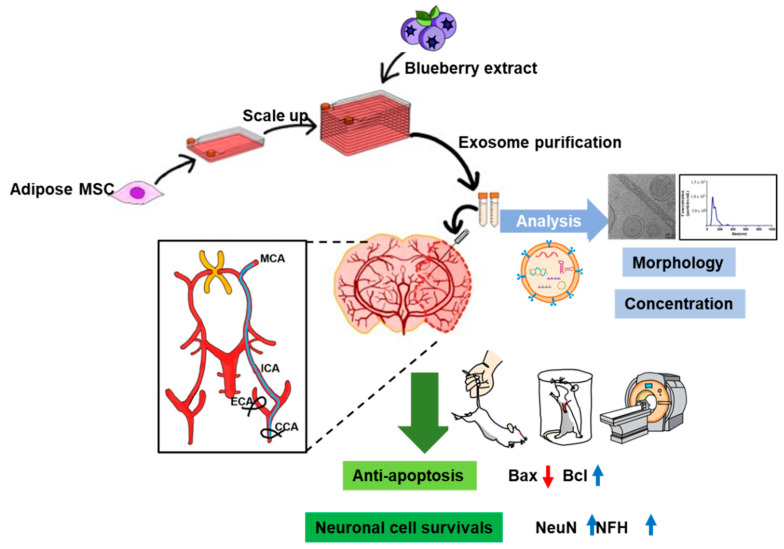
Schematic illustration of the hypothesis. We isolated the adipose tissue-derive mesenchymal stem cells from the human and scaled up the cell population with blueberry extract. The EVs were isolated and characterized by morphology and concentration via western blotting analysis, TEM (Scale bar indicates 50 nm), and NTA analysis. To investigate the effect of extracellular vesicles in ischemic conditions, we used OGD in cells and MCAo in animals. Finally, we determined that the MSC-derived extracellular vesicles with blueberry extract have therapeutic potential for regulating apoptotic signaling and increasing neuronal cell survival in in vitro and in vivo ischemic strokes. Black arrows indicate the flow of experiment. Red arrow indicates a down-regulated expression of protein. Blue arrows indicates an up-regulated expression of proteins.

**Table 1 ijms-25-06362-t001:** Concentration and size of the EVs via NTA.

Sample	Size (nm)	Number (Particles/mL)
A-EV	98.0 ± 3.3	1.74 × 10^8^ ± 5.35 × 10^6^
B-EV	102.5 ± 9.5	6.45 × 10^8^ ± 1.37 × 10^7^

**Table 2 ijms-25-06362-t002:** Gene expression profiling of A-EVs and B-EVs.

Gene	Protein Name	Foldchange	A-EV (log2)	B-EV (log2)
*SNORA23*	small nucleolar RNA, H/ACA box 23	21.153	5.394	9.797
*SNORA68*	small nucleolar RNA, H/ACA box 68	10.386	3.045	6.422
*SNORA2C*	small nucleolar RNA, H/ACA box 2C	8.662	5.402	8.516
*RN7SK*	RNA, 7SK small nuclear	8.273	3.967	7.016
*THBS1*	thrombospondin 1	6.969	3.097	5.898
*SNORD13*	small nucleolar RNA, C/D box 13	6.879	2.785	5.567
*ND5*	NADH dehydrogenase, subunit 5 (complex I)	6.726	5.067	7.817
*CYTB*	cytochrome b	6.547	5.407	8.118
*SNORA12*	small nucleolar RNA, H/ACA box 12	6.454	3.236	5.926
*SNORD89*	small nucleolar RNA, C/D box 89	6.313	2.061	4.719
*SNORD101*	small nucleolar RNA, C/D box 101	6.238	1.658	4.299
*ND6*	NADH dehydrogenase, subunit 6 (complex I)	5.847	3.576	6.124
*ND4L*	NADH dehydrogenase, subunit 4L (complex I)	5.809	6.866	9.405
*RNU4ATAC*	RNA, U4atac small nuclear (U12-dependent splicing)	5.774	1.297	3.827
*ATP8*	ATP synthase F0 subunit 8	5.681	9.069	11.575
*RPS2P46*	ribosomal protein S2 pseudogene 46	5.449	2.226	4.672
*RMRP*	RNA component of mitochondrial RNA processing endoribonuclease	5.438	6.920	9.363
*ND1*	NADH dehydrogenase, subunit 1 (complex I)	5.389	6.495	8.925
*COX1*	cytochrome c oxidase subunit I	4.989	8.334	10.653
*SNORA71D*	small nucleolar RNA, H/ACA box 71D	4.926	2.965	5.265
*SNORA80E*	small nucleolar RNA, H/ACA box 80E	4.835	5.125	7.398
*SCARNA4*	small Cajal body-specific RNA 4	4.641	5.184	7.399
*HIST1H4C*	histone cluster 1, H4c	4.418	2.224	4.368
*SBDS*	Shwachman–Bodian–Diamond syndrome	4.298	3.007	5.110
*MIR4756*	microRNA 4756	4.222	2.142	4.220
*IGKV3-11*	immunoglobulin kappa variable 3-11	4.127	1.932	3.977
*SNORA74A*	small nucleolar RNA, H/ACA box 74A	4.077	2.375	4.403
*COX2*	cytochrome c oxidase subunit II	4.005	4.824	6.826
*SNORD114-14*	small nucleolar RNA, C/D box 114-14	3.878	2.254	4.209
*FN1*	fibronectin 1	3.869	4.649	6.601
*NEAT1*	nuclear paraspeckle assembly transcript 1 (non-protein coding)	3.837	1.795	3.736
*ND2*	MTND2	3.819	5.716	7.650
*SNORD42A*	small nucleolar RNA, C/D box 42A	3.801	1.685	3.611
*TUBB4B*	tubulin, beta 4B class IVb	3.727	4.993	6.891
*LIMA1*	LIM domain and actin binding 1	3.704	2.041	3.930
*BASP1*	brain abundant, membrane attached signal protein 1	3.652	1.939	3.808
*OAZ1*	ornithine decarboxylase antizyme 1	3.642	5.324	7.189
*SNORA11*	small nucleolar RNA, H/ACA box 11	3.545	4.176	6.001
*SNORD13P3*	small nucleolar RNA, C/D box 13 pseudogene 3	3.467	3.893	5.686
*SNORD8*	small nucleolar RNA, C/D box 8	3.376	2.311	4.066
*LOC100506596*	fibril-forming collagen alpha chain	3.320	2.137	3.868
*MIR3619*	microRNA 3619	3.263	3.236	4.942
*ND3*	NADH dehydrogenase, subunit 3 (complex I)	3.246	4.931	6.630
*PRDX1*	peroxiredoxin 1	3.230	3.993	5.684
*PFN1*	profilin 1	3.146	3.434	5.088
*NUTM2D*	NUT family member 2D	3.131	3.221	4.867
*SUMO2*	small ubiquitin-like modifier 2	3.108	4.563	6.200
*TMSB10*	thymosin beta 10	3.041	8.575	10.180
*ATAD2B*	ATPase family, AAA domain containing 2B	0.321	3.121	1.482
*SNORD111*	small nucleolar RNA, C/D box 111	0.278	3.327	1.482
*LOC105371120*	uncharacterized LOC105371120	0.255	3.985	2.013
*LOC105371120*	uncharacterized LOC105371120	0.255	3.985	2.013

**Table 3 ijms-25-06362-t003:** Enriched KEGG biological pathways related to diseases and regulatory mechanisms.

KEGG Pathways	Number of Genes	*p*-Value	Bonferroni	FDR
Metabolic pathways	37	8 × 10^−7^	1.9 × 10^−4^	1.6 × 10^−5^
Oxidative phosphorylation	18	1.1 × 10^−13^	2.6 × 10^−11^	6.6 × 10^−12^
Phagosome	13	7.3 × 10^−8^	1.8 × 10^−5^	2.2 × 10^−6^
Focal adhesion	11	4.3 × 10^−5^	1 × 10^−2^	6 × 10^−4^
Regulation of actin cytoskeleton	14	3.6 × 10^−7^	8.7 × 10^−5^	9.6 × 10^−6^
Cardiac muscle contraction	87	4 × 10^−7^	9.6 × 10^−4^	9.6 × 10^−6^
Retrograde endocannabinoid signaling	148	8.1 × 10^−4^	1.9 × 10^−1^	7.4 × 10^−3^
Olfactory transduction	448	1.3 × 10^−5^	3.3 × 10^−3^	2 × 10^−4^
Thermogenesis	231	5.2 × 10^−12^	1.3 × 10^−9^	2.4 × 10^−10^
Proteoglycans in cancer	205	5 × 10^−5^	1.1 × 10^−2^	6.3 × 10^−4^
Alzheimer disease	369	6 × 10^−12^	1.4 × 10^−9^	2.4 × 10^−10^
Parkinson disease	249	2.6 × 10^−17^	6.2 × 10^−15^	3.1 × 10^−15^
Huntington disease	306	1.9 × 10^−14^	4.6 × 10^−12^	1.5 × 10^−12^
Non-alcoholic fatty liver disease	149	4.8 × 10^−7^	1.2 × 10^−4^	1.1 × 10^−5^
Pathogenic *Escherichia coli* infection	192	7.7 × 10^−4^	1.8 × 10^−1^	7.4 × 10^−3^
Shigellosis	242	1.3 × 10^−6^	3.2 × 10^−4^	2.5 × 10^−5^
Herpes simplex virus 1 infection	491	6.4 × 10^−9^	1.5 × 10^−6^	2.2 × 10^−7^

## Data Availability

The data presented in this study are available in the article and Appendix A.

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
