# Peer review of "The Therapeutic Effects of Blueberry-Treated Stem Cell-Derived Extracellular Vesicles in Ischemic Stroke"

_ijms, 2024, doi:10.3390/ijms25126362_

Round 1

Reviewer 1 Report

Comments and Suggestions for Authors

This manuscript is an interesting study analyzing the effect of treatment with blueberry on the effect of altered nucleic acids in exosome.

The problem with the analysis of nucleic acids in exosomes is the use of microarrays and the adoption of database search results as results. Validation of the results of gene expression analysis in cells or tissues is required. In particular, the gene expression in tissue or in cells should be followed up by real-time PCR or Western blot to validate the results.

In addition, authors need to present reproducible test results on the effect of blueberry treatment on cell-derived exosome.

It is necessary to indicate between which conditions the comparisons are made in order to show statistically significant differences (Figures 5 and 6).

Also, in Figure 8, positive cells should be indicated by arrows or other means.

Comments on the Quality of English Language

English quality is Good.

Reviewer 2 Report

Comments and Suggestions for Authors

The MSC-derived extracellular vesicles with blueberry extract has a therapeutic potential with regulating of apoptotic signaling and increasing the neuronal cell survivals in vitro and in vivo ischemic stroke was observed in this study. This paper is well written and scientific sound. The author provides a integrity research design and sufficient study results. However, there still a query needs to address before publication.

Comments: 

1)     Please provide the scale bar in the Figure 1E and IF and included the description in the “Figure caption” of Figure 1.

2)     Please explain why there have multiple peaks in the Figure 1C.

Comments on the Quality of English Language

none

Reviewer 3 Report

Comments and Suggestions for Authors

The manuscript titled "The Therapeutic Effects of Stem Cell-Derived Extracellular Vesicles in Ischemic Stroke" presents a comprehensive study of the potential benefits of mesenchymal stem cell-derived extracellular vesicles (MSC-EVs), particularly those treated with blueberry extracts, in treating ischemic stroke in vitro and in vivo. The following are some suggestions for improvement aimed at enhancing clarity, depth, and scientific rigor:

  1. Specify the types of extracellular vesicles used in the study early in the abstract to avoid confusion for the reader. Mention "blueberry-treated mesenchymal stem cell-derived extracellular vesicles (B-EVs)" at the beginning of the abstract to set the context clearly, maybe also include it in the title of the article, as this seems to be an important differentiation from other similar articles.
  2. Introduce earlier the concept of EVs and their general roles in cellular communication and tissue repair to provide a smoother lead-in to their application in stroke.
  3. Include a more detailed review of previous studies that have shown the effectiveness of MSCs and their derived EVs in similar models. Discuss both the successes and the limitations of past studies to better position your study's contribution.
  4. Provide more detailed reasoning for the choice of animal model and the specifics of the ischemic model used. Justify why these particular models are appropriate for the study's aims. Clarify the dosage rationale for the EV treatments provided to the animals. Include a discussion on how these dosages were determined to be therapeutic.
  5. Expand on the blueberry extract treatment. Specify the concentration and preparation of the blueberry extract used, as this could significantly affect reproducibility and the interpretation of results. Improve the description of the EV isolation process. Elaborate on how the process ensures the purity and integrity of the EVs, which is crucial for their therapeutic efficacy.
  6. Provide more information on the statistical tests used. Justify the choice of tests and discuss any corrections for multiple comparisons, particularly in the context of extensive data sets.
  7. Discuss the variability and standard deviation in the experimental data more thoroughly to assess the robustness of the findings.
  8. Deepen the discussion on how the findings compare with existing literature. Address whether the observed effects of B-EVs are consistent with the known biological activities of components found in blueberries. Explore potential mechanisms by which Blueberry-treated EVs might exert enhanced therapeutic effects. Consider the bioactive compounds involved and their possible interactions with cellular pathways.
  9. Discuss the potential translation of these findings to clinical settings. Consider the challenges and necessary steps for moving from animal models to human trials. Address scalability and manufacturing challenges in producing MSC-derived EVs for clinical use.
  10. Clearly state the limitations of the current study, including aspects related to the model systems, the generalizability of the results, and any experimental constraints.
  11. Suggest specific future studies that could further elucidate the mechanisms of action of B-EVs or explore other potential therapeutic agents for ischemic stroke. Consider the potential for combination therapies involving B-EVs and other pharmacological agents.
Comments on the Quality of English Language

Correct typographical, grammatical, or formatting errors. Ensure consistency in terminology and definitions throughout the manuscript.

Round 2

Reviewer 1 Report

Comments and Suggestions for Authors

 I recommend "Accept in present form". Thanks.

Reviewer 3 Report

Comments and Suggestions for Authors

After reading the author's response and the revised manuscript version, I recommend acceptance for publishing in the present form.